# Interleukin-10 Protects against Ureteral Obstruction-Induced Kidney Fibrosis by Suppressing Endoplasmic Reticulum Stress and Apoptosis

**DOI:** 10.3390/ijms231810702

**Published:** 2022-09-14

**Authors:** Kyongjin Jung, Taejin Lee, Jooyoung Kim, Eongi Sung, Inhwan Song

**Affiliations:** 1Department of Anatomy, College of Medicine, Yeungnam University, 170 Hyeonchung-ro, Nam-gu, Daegu 42415, Korea; 2Smart-Aging Convergence Research Center, College of Medicine, Yeungnam University, 170 Hyeonchung-ro, Nam-gu, Daegu 42415, Korea

**Keywords:** endoplasmic reticulum stress, renal fibrosis, interleukin-10, unilateral ureteral obstruction, apoptosis

## Abstract

Fibrosis is a common final pathway of chronic kidney disease, which is a major incurable disease. Although fibrosis has an irreversible pathophysiology, the molecular and cellular mechanisms responsible remain unclear and no specific treatment is available to halt the progress of renal fibrosis. Thus, an improved understanding of the cellular mechanism involved and a novel therapeutic approach are urgently required for end-stage renal disease (ESRD). We investigated the role played by interleukin-10 (IL-10, a potent anti-inflammatory cytokine) in kidney fibrosis and the mechanisms involved using *IL-10*^−/−^ mice and TCMK-1 cells (mouse kidney tubular epithelial cell line). Endoplasmic reticulum stress (ERS), apoptosis, and fibrosis in *IL-10*^−/−^ mice were more severe than in *IL-10*^+/+^ mice after unilateral ureteral obstruction (UUO). The 4-Phenylbutyrate (an ERS inhibitor) treatment induced dramatic reductions in ERS, apoptosis, and fibrosis-associated factors in the renal tissues of *IL-10*^−/−^ mice, compared to wild-type controls after UUO. On the other hand, in cultured TCMK-1 cells, the ERS inducers (tunicamycin, thapsigargin, or brefeldin A) enhanced the expressions of proapoptotic and profibrotic factors, though these effects were mitigated by IL-10. These results were supported by the observation that IL-10 siRNA transfection aggravated tunicamycin-induced CHOP and a-SMA expressions in TCMK-1 cells. We conclude that the anti-fibrotic effects of IL-10 were attributable to the inhibition of ERS-mediated apoptosis and believe that the results of this study improve the understanding of the cellular mechanism responsible for fibrosis and aid in the development of novel therapeutic approaches.

## 1. Introduction

Chronic kidney disease (CKD) has a devastating impact on quality of life. CKD progresses to end-stage renal disease (ESRD), which requires kidney transplantation or hemodialysis for survival. CKD develops as a result of chemical injury, metabolic disease, hemodynamic stress, or urinary track obstruction. Regardless of initial causes, fibrosis, and inflammation are the common pathways of CKD and ESRD. Although fibrosis is an irreversible pathophysiology, the molecular and cellular mechanisms involved remain unclear, and no specific treatment has been developed to stop the progression of renal fibrosis. Thus, an improved understanding of the cellular mechanism responsible for renal fibrosis and a novel therapeutic approach are urgently required to prevent ESRD [1].

Endoplasmic reticulum (ER) is responsible for the folding, trafficking, and modification of proteins. ER stress (ERS) is caused by the accumulation of unfolded or misfolded proteins in ER, which activates the unfolding protein response (UPR) and promotes cell death or survival [2,3]. Imbalanced ER homeostasis upregulates several genes related to kidney fibrosis, such as transforming growth factor-β (TGF-β), interleukin-10 (IL-10), IL-1β, and α-smooth muscle actin (α-SMA) [4,5]. Furthermore, chronic stress triggers epithelial cell apoptosis and the myofibroblastic differentiation of pericytes and resident fibroblasts, which lead to fibrosis [6] characterized by extracellular matrix accumulation, glomerulosclerosis, capillary and urinary space reduction, architecture distortion, vascular stability loss, and angiogenesis [7]. Considering the irreversible nature of CKD, developmental efforts should focus on the prevention of interstitial fibrosis and tubule cell apoptosis [8,9]. 

The IL-10 gene is upregulated during ERS [10], and recent studies have demonstrated that IL-10 alleviates fibrosis in various organs, including kidneys, liver, heart, and lungs, by inhibiting inflammatory responses [11]. In addition, it has been reported that IL-10 deficiency exacerbates renal inflammation and fibrosis after ureteral obstruction in mice [12,13]. Based on the results of previous studies, it is evident that IL-10 acts as a gatekeeper of the fibrotic/antifibrotic signalling pathway. Furthermore, ERS and IL-10 interact during the progression of renal fibrosis, though the mechanism involved has not been elucidated. 

In this study, we hypothesized that IL-10 and ERS levels are associated with kidney fibrosis. Hence, we investigated their relationships with the progression of kidney fibrosis induced by ureteral obstruction and ERS in *IL-10*^−/−^ mice and cultured kidney tubular epithelial cells, respectively.

## 2. Results

### 2.1. IL-10 Deficiency Exacerbated Kidney Fibrosis after UUO

Fibrosis is triggered by several factors, but ultimately it induces the proliferation of tubulointerstitial cells and results in collagen deposition. We evaluated the effect of IL-10 gene deficiency on UUO-induced kidney fibrosis using *IL-10*^−/−^ mice and *IL-10*^+/+^
*wild-type* littermates. UUO enhanced the atrophy of parenchyma, tubular injury, inflammatory cells infiltration (Figure 1A,B), and the interstitial and cellular expressions of collagens in *IL-10*^+/+^ and *IL-10*^−/−^ mice after UUO, but this effect was much more pronounced in *IL-10*^−/−^ mice (Figure 1A,C–E).

### 2.2. IL-10 Deficiency Accelerated UUO-Induced ER Stress and Apoptosis in Mouse Kidneys 

The UPR pathway is composed of three parallel signalling pathways, i.e., the PKR-like ER kinase (PERK)–eukaryotic translation initiation factor (elF), inositol-requiring enzyme 1α (IRE1α)–X-box binding protein 1 (XBP1), and activating transcription factor 6α (ATF6α) pathways, via three ER resident proteins, namely PERK, IRE1, and ATF6, and others such as the caspase-3 (Cas3) activation pathway [2]. Under conditions of ERS, binding immunoglobulin protein (BiP, an ER chaperone), which is normally bound to three ER resident transmembrane proteins, dissociates from the cell surface and regulates cell signalling, proliferation, and apoptosis [14]. Furthermore, it has been reported that when ERS increases, the C/EBP-homologous protein (CHOP) expression increases and mediates apoptosis [15]. The transcriptional targets of CHOP, which include death receptor 5 (DR5), are associated with apoptosis [16]; thus, we investigated whether increased fibrosis caused by IL-10 gene deficiency is associated with ERS. 

We found that UUO increased the protein expressions of IRE1α, PERK, elF2α, BiP, CHOP, and DR5 (all markers of ERS), and that their expressions were significantly greater in *IL-10*^−/−^ mice than in *IL-10*^+/+^ mice (Figure 2). Additionally, TUNEL-positive cells and the protein expressions of poly (ADP-ribose), polymerase (PARP), Cas3, and apoptosis inducing factor (AIF), were greater in *IL-10*^−/−^ mice, whereas the expression of Bcl-2 protein (anti-apoptotic) was lower in *IL-10*^−/−^ mice than in *IL-10*^+/+^ mice (Figure 3). These results indicate that increased apoptosis due to IL-10 gene deficiency might be associated with increased ERS. 

### 2.3. 4-PBA Halted UUO-Induced Apoptosis and Fibrosis in IL-10 Deficient Mice

The effects of IL-10 on ERS-induced apoptosis and fibrosis were confirmed by administering 4-phenyl butyric acid (4-PBA, an ERS inhibitor) to mice. After administration, *IL-10*^−/−^ mice showed dramatic increases in proapoptotic factors (BiP, PARP, and Cas3) and profibrotic factors (α-SMA and Col-1), but these increases were significantly lower in 4-PBA treated mice (Figure 4). These results indicate that UUO-induced kidney fibrosis is associated with ERS and higher kidney fibrosis levels in *IL-10*^−/−^ mice, and that this was associated with increased ERS and apoptosis. 

### 2.4. IL-10 Attenuated ERS-Induced Kidney Tubular Cell Apoptosis and the Expressions of Profibrotic Factors

Recently, it was reported that epithelial and endothelial injuries activate mesenchymal transition, inflammation, alter cytokine secretions, and result in the over-deposition of extracellular matrix in the interstitium [9,17]. We investigated the effects of IL-10 on ERS in TCMK-1 cells by treating cells with the ERS inducer, namely TG, TM, or BFG with or without IL-10. The results obtained show that IL-10 reduced the ERS-induced expressions of BiP, CHOP, and PARP (Figure 5A) and also reduced the number of TM-induced fibrosis-associated proteins, i.e., α-SMA, type I collagen (Col-1) and FN (Figure 5B). These results show that the effects of ERS on TCMK-1 damage and mesenchymal transition were significantly mitigated by IL-10 supplementation.

Finally, we confirmed the effect of IL-10 on ERS-induced apoptosis and fibrosis induction by transfecting TCMK-1 cells with IL-10 siRNA. IL-10 knockdown by siRNA transfection increased CHOP and α-SMA protein levels (Figure 5C). These results support our in vivo observations that IL-10 reduces ERS-mediated apoptosis and fibrosis.

## 3. Discussion

IL-10 is an anti-inflammatory cytokine produced by several activated immune cell types, such as T-helper (Th2) cells, NK cells, dendritic cells, macrophages, and monocytes [18]. In kidneys, IL-10 is secreted primarily by mesangial and endothelial cells [5], and several studies have reported that elevated IL-10 expression is associated with various kidney diseases, such as mesangioproliferative glomerulonephritis, IgA nephropathy, and diabetic nephropathy [19,20,21]. Furthermore, IL-10 treatment inhibited spontaneous systemic lupus erythematosus-induced kidney damage by inhibiting the Th1 response [22]. Moreover, the evidence indicates that IL-10 inhibits renal fibrosis; for example, IL-10 overexpression reduced the infiltration of inflammatory cells and mRNA in remnant kidneys of 5/6 nephrectomised rats [23], whereas an IL-10 deficiency exacerbated inflammation and fibrosis in UUO-induced mouse kidneys [24]. 

IL-10 knockout mice provide useful disease models; however, to better mimic actual pathologic conditions, alternative triggers are needed to increase IL-10 deficiency. For example, dexamethasone feeding is used to induce inflammatory bowel disease [25], and UUO to induce ERS and finally renal fibrosis [24]. In line with previous studies, we found that IL-10 deficiency significantly aggravated renal fibrosis after UUO (Figure 1) and that UUO increased IL-10 expression in the tubulointerstitium and apical regions of tubules (data not shown), as has been previously reported [26], which probably acts as a protective response to UUO [27]. Therefore, we believe that aggravated fibrosis in IL-10-deficient mice was due to the prevention of increases in IL-10 expression following UUO, leading to greater ERS, apoptosis, and fibrosis progression. However, how UUO stimulates IL-10 expression in kidneys remains a topic for future studies. 

The three UPR sensor proteins (IRE1, PERK, and ATF6) in ER determine the fate of the cells by controlling the levels of misfolded proteins [28]. Under normal conditions, these three UPR sensors associate with BiP to halt the UPR signalling pathways. However, the accumulation of misfolded proteins in the ER lumen causes BiP to separate from UPR sensor proteins and activate it. IRE1 is an ER transmembrane protein of the cytosolic kinase/endoribonuclease (RNase) domain. In the presence of ERS, activated IRE1 contributes toward life or death cell fates but is gradually inactivated. On the other hand, PERK continues its action in chronic ERS [29]. PERK kinase recognizes and phosphorylates eIF2α, and thus attenuates global protein translation and regulates cell fate, but predominantly induces apoptosis [14,29]. CHOP can induce the transcriptional activations of genes that contribute to cell death, such as DR5 a caspase-activating cell-surface death receptor that induces cell death by binding to its ligand [30,31]. We observed that levels of the major ERS sensor proteins, IRE1 and PERK, were higher in the kidney tissues of *IL-10*^−/−^ mice than in those of *IL-10*^+/+^ mice after UUO, and that BiP exhibited the same pattern. PERK, eIF2α, CHOP, and DR5 are components of the PERK-elF signalling pathway, and their expressions were also higher in *IL-10*^−/−^ mice after UUO (Figure 2). In addition, proapoptotic PARP, Cas3, and AIF levels and TUNEL-positive cell numbers were significantly higher in *IL-10*^−/−^ mice, whereas anti-apoptotic Bcl-2 levels were lower (Figure 3). This result shows that IL-10 deficiency is closely related with ERS and ERS-induced apoptosis and fibrosis. 4-PBA has been used clinically to treat urea cycle disorders and is a known inhibitor of ER stress [32,33]. 4-PBA reportedly affected intracellular UPR by directly targeting misfolded or mutant proteins [34,35]. Furthermore, several molecules, including PERK, BiP, CHOP, and c-JNK are involved in the underlying mechanisms [36]. We designed a comparable experiment by administering 4-PBA to *IL-10*^−/−^ mice. As was expected, we found that the effects of IL-10 deficiency were significantly diminished by 4-PBA (Figure 4). These results indicate that excessive ERS acts to induce kidney fibrosis [37,38], and that the anti-fibrotic effect of IL-10 is associated with ERS reduction.

Previous studies have reported that prolonged or excessive ERS and perturbed ERS homeostasis aggravates renal fibrosis. For example, CHOP deficiency inhibited fibrosis via Hmgb1/TLR4/NFκB signalling through a CHOP-related ERS pathway [39]; ERS caused tubular damage in kidney disease [40,41]; and irreversible ERS caused kidney cell apoptosis and, consequently, fibrosis [3,42]. Fibrosis is the final result of tissue damage and subsequent inflammation. Damaged cells induce inflammation by increasing leukocyte infiltration, and cytokines and oxidative radicals are released with leukocytes [43]. This is followed by cell proliferation, regeneration, extracellular matrix production, and eventually scar formation [43]. Inflammation and oxidative stress are common features of many kidney diseases. Recently, biomarkers for kidney injuries have focused on oxidative stress [44] and inflammation [45]. In addition, renal fibrosis develops as a result of epithelial-to-mesenchymal transition induced by cell damage. In particular, tubular epithelial cell death dramatically induces this cascade [6,46]. Moreover, ERS was recently found to be a factor of renal fibrosis [17,47]

Several antibiotic or antineoplastic chemicals can induce ERS. BFA induces ERS by inhibiting protein transport from the ER to the Golgi apparatus [48], TG by blocking N-linked glycosylation in ER, and thus disrupting protein maturation [49] and TM by inhibiting the ER Ca^2+^ ATPase pump [50,51]. We tested the effects of ERS directly by treating TCMK-1 cells with the abovementioned ERS inducers. These treatments increased the expressions of proapoptotic and profibrotic proteins, but these increases were mitigated by IL-10 supplementation (Figure 5A,B); furthermore, these effects were opposite to those induced by IL-10 siRNA treatment (Figure 5C).

Our results indicate that the anti-fibrotic effects of IL-10 are due to the inhibition of ERS-mediated apoptosis. We believe the presented results contribute to an increased understanding of the topic and encourage further studies on the cellular mechanisms involved and on the development of novel therapeutic approaches to fibrosis.

## 4. Materials and Methods

### 4.1. Animal Preparation

The experiments were performed using 10-week-old male *IL-10*^−/−^ and *IL-10*^+/+^ mice, purchased from the Jackson Lab (Bar Harbor, ME, USA) and the Hana Company (Busan, Korea), respectively. Mice were provided free access to water and standard mouse chow. Each animal group consisted of at least three mice. Animals were anesthetized with 0.02 mL/g at a body weight of 1.25% Avertin (Sigma, St. Louis, MO, USA) before surgery. To induce UUO, the left kidney was exposed through a left-flank incision, and the left ureter was then completely obstructed near the renal pelvis using a 6/0 nylon tie. Sham operations were performed using the same surgical procedure without ligation. To assess the effect of IL-10 on ERS, animals were treated with 4-PBA (100 mg/kg BW; Sigma) 3 times daily on alternate days for 7 days from the day after surgery. Kidneys were harvested 7 days after surgery and subjected to histologic, biochemical, and Western blot analyses.

Animal studies were approved beforehand by the Institution Animal Care and Use Committee of Yeungnam University College of Medicine, and all procedures were performed in accordance with the guidelines.

### 4.2. Cell Culture and Treatment

TCMK-1 cells (a mouse kidney tubular epithelial cell) were provided by Tae Jin Lee, Yeungnam University, Daegu, Korea and cultured in high glucose-DMEM supplemented with 10% FBS and antibiotics in a 5% (*v*/*v*) CO_2_-humidified atmosphere at 37 °C. Confluent (70–80%) cells were incubated for 15–18 h in serum-free medium and treated with IL-10 (10 ng/mL; PeproTech Inc., Rocky Hill, NJ, USA), tunicamycin (TM; 0.1 μg/mL; Sigma), thapsigargin (TG; 0.1 μM; Sigma), or brefeldin A (BFA; 10 μM; Sigma) for 20 h. The cells were then subjected to IL-10 siRNA transfection and protein expression analyses for ERS or fibrosis-related proteins.

### 4.3. Small Interfering RNA (siRNA) Treatment In Vitro

The IL-10 and control siRNA duplexes used in this study to transiently deplete IL-10 in TCMK-1 cells were purchased from the Bioneer Corporation (Daejeon, Korea). TCMK-1 cells were cultured until they were 50–60% confluent and were then transfected with siRNA oligonucleotides using Lipofectamine 2000 (Invitrogen; Carlsbad, CA, USA), according to the manufacturer’s recommendations. After 4 h of transfection, cells were maintained in fresh medium-containing serum for 20 h, and then a Western blot analysis was performed.

### 4.4. Western Blot Analysis

For the Western blotting experiments, protein electrophoresis was performed as described previously [52]. Proteins were transferred to Immobilon-P membranes (Millipore, Bedford, MA, USA), blocked with 5% skimmed milk in TBST, and incubated overnight at room temperature using antibodies against the following: IL-10 (1:1000; Abcam, Cambridge, MA, USA), α-SMA (1:10,000; Sigma); Col-1 (1:1000; Abcam); fibronectin (FN; 1:1000; BD Biosciences, Franklin Lakes, NJ, USA); cleaved Cas3 (1:1000; LS Bio, Seattle, WA, USA); PARP (1:1000; Cell Signalling, Danvers, MA, USA); Bcl2 (1:1000; Santa Cruz Biotechnology, Santa Cruz, CA, USA); Bcl-xL (1:1000; Santa Cruz); DR5 (1:1000; KOMA Biotech, Seoul, Korea); AIF (1:1000; Cell signalling); BiP (1:1000; Cell Signalling); CHOP (1:1000; Cell Signalling); IRE1 (1:1000; Cell Signalling); PERK (1:1000; Cell Signalling); and eIF2α (1:1000; Cell Signalling). After washing, membranes were incubated for 1 h at room temperature with a horseradish peroxidase conjugated anti-mouse or anti-rabbit antibody (1:5000; Santa Cruz). Specific proteins were visualized by enhanced chemiluminescence (PerkinElmer, Waltham, MA, USA). GAPDH (1:5000; Santa Cruz) and β-actin (1:10,000; Santa Cruz) were used as a loading control and for normalizing immunoblots, respectively, which were analyzed using NIH Image J.

### 4.5. Immunohistochemical Analysis

Kidney tissues were fixed in 3.7% paraformaldehyde, embedded in paraffin, and cut into 4 μm sections using a microtome. Sections were Masson trichrome and periodic acid-Schiff (PAS) stained using a standard protocol, and five fields (0.1 mm^2^/field) of kidney cortices per section were taken using an optical microscope (Leica CTR 6000, Germany) and analyzed using i-Solution software (IMT i-Solution Inc., Conquitlam, BC, Canada). Damage was recorded as previously described [52]. Briefly, damage levels were scored as follows: no damage (0 points); mild damage, defined as a dilated tubular lumen with flattened epithelial cells (1 point); moderate damage includes congestion of tubule as well as the factors of mild damage (2 points); and severe damage as indicated by tubule destruction and flattened epithelial cells without nuclear staining (3 points). Damage score indices were calculated by dividing the summed cell number score products by the total numbers of tubules.

### 4.6. TUNEL Assay Analysis

Sections were deparaffinized, permeabilized with 0.2% Triton X-100 for 10 min, blocked with 2% BSA/PBS for 30 min, and washed with PBS. TUNEL cell death assays were conducted according to the manufacturer’s instructions (Roche Molecular Biochemicals, Mannheim, Germany). Cell nuclei were counterstained with 10 µg/mL 4′,6-diamidino-2-phenylindole (DAPI, Sigma, USA), and slides were mounted with Vectashield (Vector Laboratories Inc., Burlingame, CA, USA). Images of cultured cells in dishes were also captured using an inverted fluorescence microscope.

### 4.7. Statistical Analysis

The significances of the differences between groups were determined using a one-way analysis of variance (ANOVA) and Tukey’s post-hoc test utilizing the Prism software. Results are presented as means ± standard errors, and significance was determined with a *p* value < 0.05. In all cases, groups consisted of at least three mice.

## Figures and Tables

**Figure 1 ijms-23-10702-f001:**
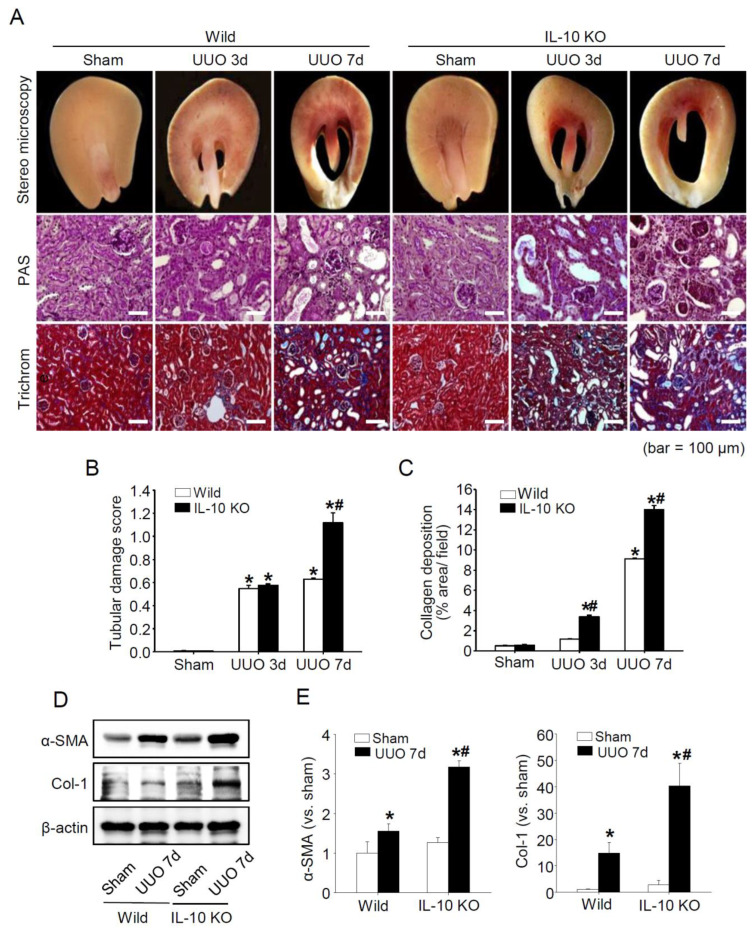
Representative histological changes and fibrosis-related protein expressions in kidneys after UUO. Gross and microscopic findings of kidneys after UUO (**A**). Structural disturbance was evaluated with a Tubular damage score and expressed as a graph (**B**). Collagen deposition was evaluated by quantifying Masson’s trichrome stained areas using Image-Pro Plus software (**C**). Seven days after UUO injury, the expressions of α-SMA and Col-1 were determined with Western blotting. β-actin was used as the loading control. Densities of bands were quantified using ImageJ software (*n* = 3 to 4) and bars indicate means ± SEMs (**D**,**E**). * *p* < 0.05 vs. *IL-10*^+/+^-sham mice; # *p* < 0.05 vs. UUO induced *IL-10*^+/+^ mice. PAS and Trichrome stain. Scale bars represent 100 μm.

**Figure 2 ijms-23-10702-f002:**
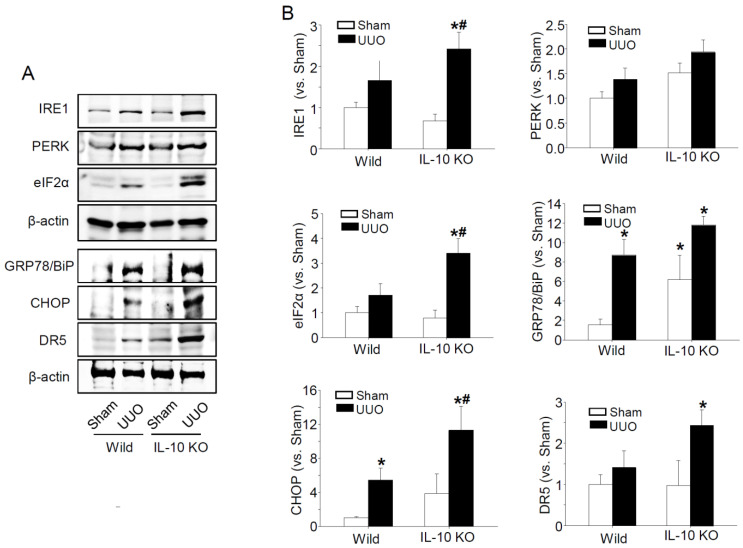
Expressions of ER stress-related proteins in kidneys of the *IL-10*^−/−^ and control wild mice 7 days after UUO. ERS signalling pathway protein levels (IRE1, PERK, eIF2α, BiP, CHOP, and DR5) were determined with Western blotting (**A**). β-actin was used as a loading control. Densities of bands were quantified using ImageJ software (*n* = 3 to 4) and bars indicate means ± SEMs (**B**). * *p* < 0.05 vs. *IL-10^+/+^*-sham mice, # *p* < 0.05 vs. UUO induced *IL-10*^+/+^.

**Figure 3 ijms-23-10702-f003:**
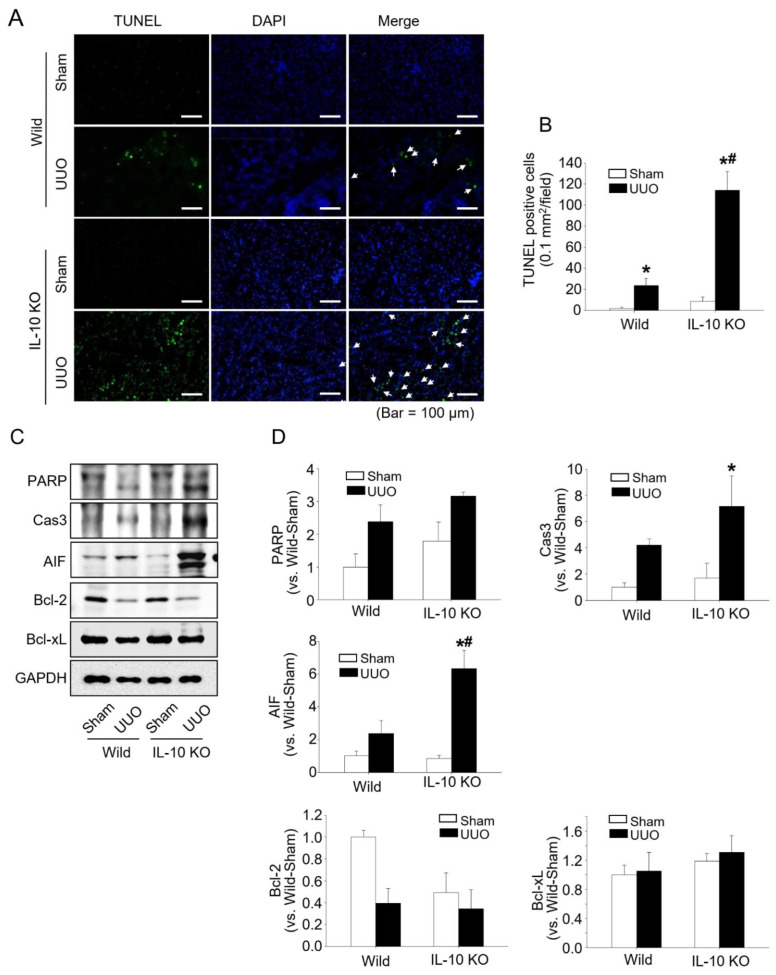
IL-10 deficiency aggravated apoptosis in the kidney 7 days after UUO. Apoptotic cells after UUO were detected by TUNEL staining (green and arrows pointed in merged figures) kidney tissues and were significantly more numerous in *IL-10*^−/−^ mice than in *IL-10*^+/+^. Nuclei were stained with DAPI (blue) (**A**). Histogram obtained by counting cells stained with TUNEL and DAPI (**B**). PARP, Cas3, AIF, Bcl-2, and Bcl-xL levels were determined with Western blotting. GAPDH was used as the loading control. Densities of bands were quantified using ImageJ software (*n* = 3 to 4) and bars indicate means ± SEMs (**C**,**D**). * *p* < 0.05 vs. *IL-10*^+/+^-sham mice; # *p* < 0.05 vs. UUO induced *IL-10*^+/+^ mice.

**Figure 4 ijms-23-10702-f004:**
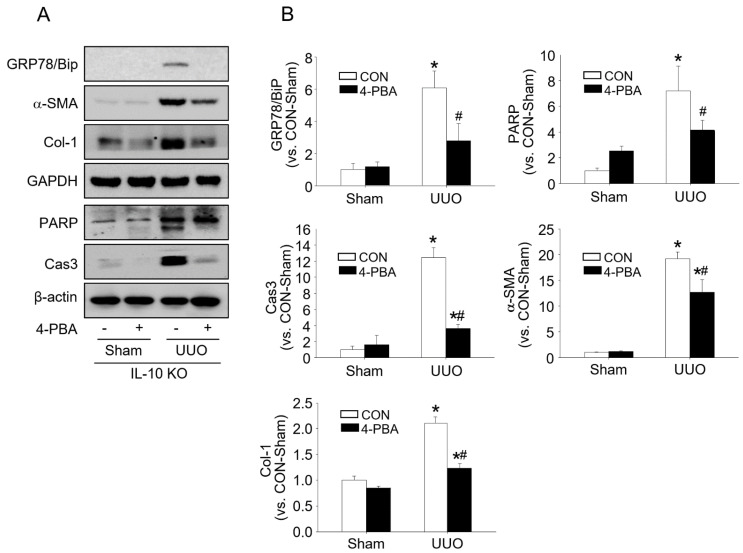
4-PBA treatment inhibited ER stress, fibrosis, and apoptosis 7 days after UUO in *IL-10*^−/−^ mice. ERS-related protein (BiP), apoptosis-related protein (PARP and Cas3), and fibrosis-related protein (α-SMA and Col-1) levels in 4-PBA or control saline treated *IL-10*^−/−^ mice were determined with Western blotting (**A**). GAPDH was used as the loading control. Densities of bands were quantified using ImageJ software (*n* = 3 to 4) and bars indicate means ± SEMs (**B**). Results are presented as means ± SEMs (*n* = 3 to 4) * *p* < 0.05 vs. *IL-10*^−/−^-sham mice not administered 4-PBA, # *p* < 0.05 vs UUO induced CON mice.

**Figure 5 ijms-23-10702-f005:**
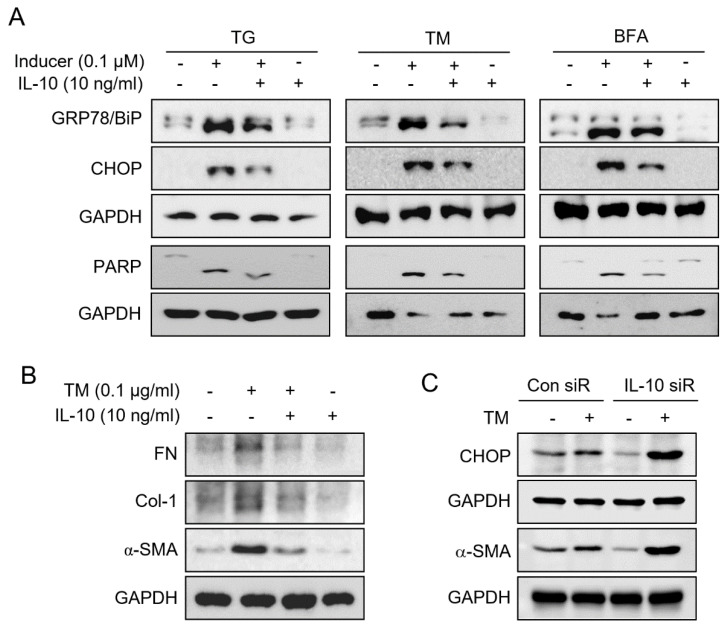
IL-10 impeded ER stress and fibrosis in tubular epithelial cells. BiP, CHOP, and PARP levels in ERS inducers (TG, TM and BFA) treated TCMK-1 cells with or without IL-10 pre-expose were determined by Western blotting (**A**). Fibrosis-related protein levels were also determined in TM treated TCMK-1 cells with or without IL-10 pre-expose (**B**). CHOP and α-SMA levels in TM treated TCMK-1 cells were determined by Western blotting after transfected with by IL-10 siRNA or Con siRNA (**C**). GAPDH was used as the loading control.

## Data Availability

Not applicable.

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
