# Peer review of "Interleukin-10 Protects against Ureteral Obstruction-Induced Kidney Fibrosis by Suppressing Endoplasmic Reticulum Stress and Apoptosis"

_ijms, 2022, doi:10.3390/ijms231810702_

Round 1
Reviewer 1 Report
The Authors proposed an analysis of Interleukin-10 protection in a model of ureteral obstruction-induced kidney fibrosis. In my opinion, this article lacks novelty, and some data raised important questions.
Major points
- The role of IL-10 deficiency in unilateral ureteral obstruction in mice is well detailed in Jin et al. (Ref#25). This article only confirmed these data, and the evidence of endothelial stress reticulum does not provide, in my opinion, sufficient novelty to the actual results presented in the manuscript. Based on the evidence of IL-10 involvement in several diseases (e.g., AKI, see Wei et al. 2022, doi 10.1016/j.intimp.2022), I suggest considering other conditions or, at least, including similar experiments on human tubular cells to assess the practical importance of this mechanism in human diseases.
- The notes for every figure are not self-explicative and should be expanded; I also have some concerns about the quantified bar densities in some figures (i.e., in Figure 3, panel C PARP and CAS3 are challenging to discriminate, as for CHOP in Figure 4 Panel A; in AIF two bars are notable in IL-10 KO UUO). The Authors are invited to improve the quality of all the figures, even adding additional experiments in cases of difficult quantification.
Minor questions
- Considering that the paper is focused on an animal model, this information should be stressed in the title
- Add a graphical scheme of the adopted experimental protocol
Author Response
Thank you very much for the kind review and also for the detailed recommendation. We agree with all your recommendations, but we list our views as follows:
Major points
- The role of IL-10 deficiency in unilateral ureteral obstruction in mice is well detailed in Jin et al. (Ref#25). This article only confirmed these data, and the evidence of endothelial stress reticulum does not provide, in my opinion, sufficient novelty to the actual results presented in the manuscript. Based on the evidence of IL-10 involvement in several diseases (e.g., AKI, see Wei et al. 2022, doi 10.1016/j.intimp.2022),
â–º We agree that the animal model and the main subject of this study are the same as other studies (Ref#25). Knockout animal models are a useful tool for in vivo research and are applied to the study of a number of diseases. We also used IL-10 KO mice for the study of other diseases, such as inflammatory bowel disease (https://doi.org/10.5009/gnl18438, 10.1016/j.tice.2014.12.001, https://doi.org/10.5056/jnm14008). Fibrosis is the most frequent and extensively studied topic in kidney disease.
We believe that the 'fibrosis study with IL-10 KO mice' may be one of the common test formats for kidney studies. Jin et al. (Ref#25) only reported tissue fibrosis using IL-10 KO mice, but we investigated the ER stress mechanism, one of the cutting-edge topic in cell physiology.
The PubMed search has 126 papers for the keywords "ER stress and kidney fibrosis" and 163 papers for "IL-10 and kidney fibrosis", but there is 0 paper searched for "IL-10, renal fibrosis and ER stress".
This manuscript may be the first to report the effects of IL-10 in ER stress-induced renal fibrosis using IL-10 KO mice. Please confirm that you did not confuse endoplasmic reticulum stress (ERS) with endothelial stress reticulum.
I suggest considering other conditions or, at least, including similar experiments on human tubular cells to assess the practical importance of this mechanism in human diseases.
â–º Of cause, the most important and accurate data for human diseases are data obtained from humans. And the results of animal studies take precedence over those of the cells in the dish. However, with the rapid expansion of experimental ethical issues controlled by the IRB and IACUC, obtaining data simply from humans or materials of human origin is becoming increasingly limited, and is also driving laboratory research rather than research using living animals.
We believe that, given the circumstances, an effective combination of in vivo and in vitro experiments and human results may be ideal for reaching the goal and finally to the bedside. Also for us, this is one of the important topics I always discuss with others as a researcher, as a doctor, and as the chair of our IACUC.
- The notes for every figure are not self-explicative and should be expanded; I also have some concerns about the quantified bar densities in some figures (i.e., in Figure 3, panel C PARP and CAS3 are challenging to discriminate, as for CHOP in Figure 4 Panel A; in AIF two bars are notable in IL-10 KO UUO). The Authors are invited to improve the quality of all the figures, even adding additional experiments in cases of difficult quantification.
â–º Notes on the resulting data of UUO mice and sham operation mice have been expanded. As you pointed out, the figures are not so clean but please understand that Western blotting has sometimes not given clean images as you know. We hope that you understand the circumstances in which the experiment cannot be repeated because the animals used in the experiment cannot be raised at present. We again measured the density of the Cleaved-PARP form in Fig. 3C, and the newly revised Fig. 3D.
We totally agree with the reviewer's comments. We hope that you understand the circumstances in which the experiment cannot be repeated because the animals used in the experiment cannot be raised at present. Figure 4 The CHOP data of Panel A was removed from the drawing and also from the manuscript.
Minor questions
- Considering that the paper is focused on an animal model, this information should be stressed in the title
â–º Although the in vivo experiments is larger than the in vitro experiments, the in vitro data include ERS inhibition studies as well as IL-10 siRNA assays, which could be the culmination of this study. Please understand that including more details in the title can make the title too long and confusing.
- Add a graphical scheme of the adopted experimental protocol
â–º We tried a graphical scheme, but it was not harmonious. This is a little bit large and complex study, so to aid the reader’s understanding, we simply group the results by subject and add a short description at the beginning of each result group.
Reviewer 2 Report
1) Figure 1: Time course data of SCr and BUN should be added.
2) The authors should add a reference () in the "Damaged cells induce inflammation by increasing leukocyte infiltration, and with leulpcyte release cytokines and oxidative radicals" (Page 8, Lines224-226).
3) In the Discussion section, 4th paragraph is too short. Then, description about inflammation should be added. Then, the following descriptions had better be added: "Inflammation and oxidative stress are common features of many kidney diseases. Recently, biomarkers for kidney injuries focused on oxidative stress (Toxicology. 2011 Nov 28;290(1):82-8.) and inflammation (Kidney Int Rep. 2022 Apr 5;7(7):1514-1523.)" after "This is followed by cell proliferation, regeneration, and extramatrix production, eventually scar formation [44]. (page 8, Lines226-227).
3) Abbreviation should be describe at first appearance. (e.g. SHOP (page 2, Line85 is spelled out in the Methods section (Page 10, Line 293); DR5 (page 2, line 86) is spelled out in the Discussion section (Page 8, Line 201).
4) Figure 2B: In the figure for eIF2a, "Sha" is incorrect. Is it "Sham"?
Author Response
Thank you very much for the kind review and also for the detailed recommendation. We agree with all your recommendations, but we list our views as follows
1) Figure 1: Time course data of SCr and BUN should be added.
â–º We agree that BUN and creatinine are one of the most important indicators of renal function. Because the purpose of this study is the mechanism of ERS induced renal fibrosis, we could not include functional analysis to focus more on the ERS mechanism.
2) The authors should add a reference () in the "Damaged cells induce inflammation by increasing leukocyte infiltration, and with leukocyte release cytokines and oxidative radicals" (Page 8, Lines224-226).
â–º Reference is inserted
3) In the Discussion section, 4th paragraph is too short. Then, description about inflammation should be added. Then, the following descriptions had better be added: "Inflammation and oxidative stress are common features of many kidney diseases. Recently, biomarkers for kidney injuries focused on oxidative stress (Toxicology. 2011 Nov 28;290(1):82-8.) and inflammation (Kidney Int Rep. 2022 Apr 5;7(7):1514-1523.)" after "This is followed by cell proliferation, regeneration, and extra matrix production, eventually scar formation [44]. (page 8, Lines226-227).
â–º Changed as recommend
3) Abbreviation should be describe at first appearance. (e.g. SHOP (page 2, Line85 is spelled out in the Methods section (Page 10, Line 293); DR5 (page 2, line 86) is spelled out in the Discussion section (Page 8, Line 201).
â–ºAll cases are modified as directed.
4) Figure 2B: In the figure for eIF2a, "Sha" is incorrect. Is it "Sham"?
â–º We appreciate for reading in detail.
Round 2
Reviewer 1 Report
The authors revised the presented results without additional experiments, improving the quality of the paper.